# C-C Motif Chemokine Ligand 5 (CCL5) Promotes Irradiation-Evoked Osteoclastogenesis

**DOI:** 10.3390/ijms242216168

**Published:** 2023-11-10

**Authors:** Jing Wang, Fanyu Zhao, Linshan Xu, Jianping Wang, Jianglong Zhai, Li Ren, Guoying Zhu

**Affiliations:** Department of Radiological Hygiene, Institute of Radiation Medicine, Fudan University, 2094 Xietu Road, Shanghai 200032, China; 21211140004@m.fudan.edu.cn (J.W.); 22211140009@m.fudan.edu.cn (F.Z.); 19211140005@fudan.edu.cn (L.X.); jianpingwang@fudan.edu.cn (J.W.); jlzhai@fudan.edu.cn (J.Z.); 20211140001@fudan.edu.cn (L.R.)

**Keywords:** osteocyte, cellular senescence, CCL5, JAK1/STAT3, osteoclastogenesis

## Abstract

The imbalance that occurs in bone remodeling induced by irradiation (IR) is the disruption of the balance between bone formation and bone resorption. In this study, primary osteocytes (OCYs) of femoral and tibial origin were cultured and irradiated. It was observed that irradiated OCY showed extensive DNA damage, which led to the initiation of a typical phenotype of cellular senescence, including the secretion of senescence-associated secretory phenotype (SASP), especially the C-C motif chemokine ligand 5 (CCL5). In order to explore the regulation of osteoclastogenic potential by IR-induced senescent OCYs exocytosis factor CCL5, the conditioned medium (CM) of OCYs was co-cultured with RAW264.7 precursor cells. It was observed that in the irradiated OCY co-cultured group, the migration potential increased compared with the vehicle culture group, accompanied by an enhancement of typical mature OCs; the expression of the specific function of enzyme tartrate-resistant acid phosphatase (TRAP) increased; and the bone-destructive function was enhanced. However, a neutralizing antibody to CCL5 could reverse the extra-activation of osteoclastogenesis. Accordingly, the overexpression of p-STAT3 in irradiated OCY was accompanied by CCL5. It was concluded that CCL5 is a potential key molecule and the interventions targeting CCL5 could be a potential strategy for inhibiting osteoclastogenesis and restoring bone remodeling.

## 1. Introduction

As society develops, population aging is becoming a severe burden worldwide [1,2]. According to the communiqué of China’s seventh population census, the percentage of older adults has continued to rise, from 6.9 to 13.5 percent of the total population from 2000 to 2020 [3]. This is accompanied by a growing number of age-related chronic diseases, including vascular disease, diabetes, renal failure, neurodegeneration, arthritis, stroke, cancer, osteoporosis, etc. [3,4].

Cellular senescence is a degenerative process that occurs with natural aging; it is primarily divided into replicative senescence caused by telomere shortening and stress senescence caused by a variety of stressors such as oxidative stress, oncogenes, irradiation (IR), and mitochondrial dysfunction via a sustained DNA damage response [5]. Of these, IR, one of the three major means of tumor treatment at present, achieves good tumor therapeutic efficacy; however, it can also lead to imbalances in skeletal reconstruction in the targeted radiation area and even in distant unirradiated areas [6]. In addition, IR would also induce a disruption of the bone microenvironment through the disintegration of the bone matrix and the release of active factors, ultimately increasing the risk of bone metastasis [7,8,9,10,11,12,13].

Bone is a complex organ that relies on a dynamic balance between osteoblasts (OBs) and osteoclasts (OCs), as well as tight regulation of that balance by osteocytes (OCYs) [9,10]. Previous regulatory interventions for bone imbalance have mostly focused on activating OBs or inhibiting OCs. However, the unidirectional activation or inhibition of a certain type of cell function makes it difficult to tightly regulate bone metabolic homeostasis [11,12], thus necessitating the development of a new intervention strategy targeting different cells of the bone microenvironment. Among them, OCYs are the master regulators of bone remodeling, not only playing a role in the maintenance of bone structural integrity and mechanical stress support but also coordinating the balance of OB-OC coupling across cellular synapses. As long-lived cells, OCYs are prone to accumulating damage and leading to premature senescence, and they are one of the main cells that can secrete senescence-associated secretory phenotype (SASP) [13,14,15]. Thus, this study mainly focuses on the senescence of OCYs [13].

SASP contains pro-inflammatory cytokines, chemokines, extracellular matrix proteins, and growth factors, which can regulate the homeostasis of bone metabolism and are crucial in physiological/pathological bone loss [16]. The expression profile of 110+ SASP components secreted by IR-induced senescent OCYs was established in our laboratory previously using a cellular multifactor antibody microarray assay [8,17]. Based on the results of previous experiments, in conjunction with the bioinformatic analysis and literature search, in this study, we focus on the C-C motif chemokine ligand 5 (CCL5), which plays a regulatory role in osteoclastogenesis [18,19,20,21], given the fact that CCL5 promotes the migration of OC precursor cells in a concentration-dependent manner, enhancing osteoclastogenesis. Accordingly, CCL5 can not only stimulate the elevated intracellular Ca^2+^ concentrations and prolonged calcium flow, which results in hyperphosphorylation and extensive cellular activation, but it can also activate the JAK1/STAT3 pathway, which is a stressful inflammatory signaling pathway that has the ability to cascade the production and secretion of key SASP factors, especially CCL5, that regulate bone metabolism. Therefore, we hypothesized that an increase in CCL5 among the SASP factors secreted by OCYs would activate IR-induced inflammation and osteoclastogenesis through the regulation of the JAK1/STAT3 pathway.

## 2. Results

### 2.1. Radiation Impairs the Morphology and Function of OCYs

OCY is the most abundant cell in the skeleton and is also the primary sensor, integrator, and single transducer that coordinates skeletal development, maintenance, and healing. In this study, a modified sequential digestion method was used to isolate primary OCYs from the femur and tibia of BALB/c mice for in vitro culture and subsequent experiments. Meanwhile, the identification of primary OCYs was based on cell morphology with slender synapses, E11-specific protein expression, and weak ALP expression. To observe the impact of IR on OCY biological function, the OCY viability, synaptic length, and functional protein expression were examined for 3 days post-irradiation. Radiation exposure resulted in a significant reduction in the percentage of live cells in total OCYs, with statistically significant differences (Figure 1A; Appendix A). OCYs also emit long protrusions that can deliver molecules to other cells in the adjacent bone marrow microenvironment. Furthermore, the SEM and F-actin fluorescence staining of irradiated OCYs revealed the presence of degraded dendrites and disorganized cytoplasm, where there was a tendency to lose dendritic branching structures, with a 47.37% shortening of the dendrites (Figure 1A,B; Appendix A), accompanied by a statistically significant downregulation of the protein and RNA expression of the specific E11 (Figure 1B). The results of the Western blot and RT-qPCR analysis also revealed that the functional marker of RANKL expression, which activates bone resorption and inhibits bone formation, was upregulated in irradiated OCYs, whereas the expression of DMP-1 and OPG was significantly downregulated (Figure 1C). Together, these observations demonstrate that IR can disrupt the regulation of bone metabolism by causing a decrease in viability, morphologic damage, and the functional impairment of OCYs.

### 2.2. Radiation Initiates Cellular Senescence in OCYs

The induction of OCY senescence by IR was determined by the simultaneous presence of multiple markers, including senescence-associated β-galactosidase (SA-β-gal) expression, the formation of senescence-associated heterochromatic foci (SAHF), the persistent DNA damage response (DDR), and the activation of senescence-associated pathways *p53-p21^WAF1/CIP1^ (p21)* and *p16^INK4a^ (p16)-Rb.* Among these, β-galactosidase activity associated with senescence was as a surrogate marker for the increased content of lysosomal in senescent cells. The percentage of SA-β-gal^+^ cells in the irradiated group increased by almost 2.5-fold compared to the control group (Figure 2A; Appendix A). Furthermore, DAPI staining demonstrated the disruption of chromatin structure in the nucleus of irradiated OCYs, the disruption of nuclear integrity, and the formation of SAHF, which was manifested as a significant increase in the percentage of SAHF^+^ cells. In particular, 10.01% of the cells were SAHF^+^ in the control group and 27.11% in the irradiated group (Figure 2A; Appendix A). Similarly, the appearance of γ-H2AX is closely associated with DNA double-strand breaks (DSBs). γ-H2AX immunofluorescence staining showed that IR enhanced the accumulation of γ-H2AX in OCYs, and the percentage of γ-H2AX^+^ cells was increased by a factor of 3.12 in the irradiated group compared with the control group (Figure 2B; Appendix A). Additionally, the Western blot analysis also revealed a relatively high expression of the γ-H2AX protein in irradiated OCYs (Figure 2B). Furthermore, the activation of senescence-associated protein pathways p16 and p21 was detected using Western blot and RT-qPCR, and the expression of p16 and p21 was found to be highly upregulated in irradiated OCYs, with statistically significant differences (Figure 2C). Collectively, these findings suggested that IR does induce DNA damage and further stimulates OCY senescence.

### 2.3. Radiation Enhances SASP Secretion in OCYs

As a typical indicator of cellular senescence, SASP consists of multiple components such as pro-inflammatory factors, growth factors, chemokines, tissue inhibitors of metalloproteinases, etc. To determine if IR can alter SASP secretion, a Mouse XL Cytokine Array Kit (ARY028, R&D, USA) was utilized to identify the differential expression of various factors in the supernatants of irradiated OCYs. In summary, we found that the release of fourteen SASP factors was significantly increased in irradiated OCYs compared with the controls, such as MMP-9, myeloperoxidase, EGF, and CCL5, whereas there was a decrease in the release of some cytokines, such as CCL11/Eotaxin (Figure 3A). It is interesting to note that the expression of the chemokine *CCL5*, pro-inflammatory cytokines *IL-1α*, tumor necrosis factor *TNF-α*, *NF-κB*, which mediates osteoclastogenesis, and the *MMP-3*, *MMP-9*, and *MMP-13* classes, which indicate the maturation of OCs, was significantly upregulated, as verified with RT-qPCR, whereas the expression of anti-inflammatory cytokines such as interleukin *IL-4*, *IL-10*, and *IL-13* was downregulated in the irradiated OCYs, with statistically significant differences (Figure 3B). Together, these results demonstrate that in addition to direct cellular dysfunction, IR-induced senescent OCYs can develop the overexpression and secretion of SASP in the surrounding environment; these hallmarks are synovial inflammation and irreversible bone destruction.

As noted above, CCL5 in irradiated OCYs was significantly increased up to 1.28-fold compared with the control. Moreover, the ELISA results showed an increase of 13.33% in CCL5 in the supernatants of irradiated OCYs compared with the controls (Figure 3C). Interestingly, immunofluorescence staining for CCL5 showed that IR promoted the secretion of CCL5 in OCYs, with a 1.25-fold increase in the percentage of CCL5^+^ cells in the irradiated group compared with the control group (Figure 3C; Appendix A). In addition, the activation of CCL5 was detected using Western blot and RT-qPCR, and it was found that the protein and gene expression of CCL5 were strongly upregulated in irradiated OCYs with statistically significant differences (Figure 3B,C). Collectively, the results of the above experiments point to the significant upregulation of CCL5 in the IR-induced senescent OCYs, thus providing a novel insight into the senescent OCY regulation of the bone microenvironment via the paracrine secretion of CCL5.

### 2.4. CCL5 Plays an Important Role in Promoting Osteoclastogenesis

OC overactivity leads to microarchitectural defects and a low bone mass. To verify the paracrine regulation effect on osteoclastogenesis by the IR-induced chemokine CCL5 of OCYs, the CM of OCYs was collected 3 days post-irradiation with or without 0.5 μg/ml CCL5 neutralizing antibody intervention. The CM was co-cultured with RAW264.7 precursor cells. Twenty-four hours postoperatively, the wound-healing rate of the CM of the control group (CM-0 Gy) was 18.33%, compared with 22.78% when considering the CM of irradiated OCYs (CM-2 Gy), and 19.91% when considering the CM of irradiated OCYs with the addition of 0.5 μg/mL CCL5 neutralizing antibody (CM-2 Gy + anti-CCL5) (Figure 4A; Appendix A). Furthermore, the Transwell migration assay showed that the migration number of RAW264.7 precursor cells with CM-2 Gy was increased by a factor of 1.62 compared with the control group, while the neutralizing antibody to CCL5 reversed the effect, significantly decreasing the migration number of RAW264.7 precursor cells by as much as 35.30% (Figure 4A; Appendix A). TRAP is an OC-specific marker enzyme; positive TRAP staining could reflect the differentiation ability of OC. In this study, we observed a significant increase in TRAP^+^ areas in OCs with CM-2 Gy compared with the control group. It was demonstrated that the number of large and medium-sized OCs (nuclei ≥ 6) with CM-2 Gy was higher than that of the control group. Nevertheless, the neutralizing antibody to CCL5 was shown to reduce TRAP^+^ areas and inhibit the OC differentiation ability (Figure 4B; Appendix A). Simultaneously, according to the staining results of the F-actin cytoskeleton, OCs co-cultured with CM-2 Gy displayed a typical morphology of mature OCs with bands of pseudopods with smooth margins, while the neutralizing antibody to CCL5 disrupted the actin ring, and some OCs had dendritic pseudopods and exhibited relatively immature cell morphology (Figure 5A; Appendix A). The activation of OC functional genes and adhesion factors was also detected using RT-qPCR, and the expression of *TRAP*, *CTSK*, *Oscar*, *Ocstamp*, and integrin *αV*, *β3* was found to be strongly upregulated in the OC with CM-2 Gy, while the neutralizing antibody to CCL5 reversed the overexpression of these genes (Figure 5B). Taken together, the above results suggest that the overexpression of CCL5 by IR-induced senescent OCYs accelerates the migration of RAW264.7 precursor cells and promotes the OC differentiation ability. 

### 2.5. CCL5 Activates the JAK1/STAT3 Pathway

The aforementioned results demonstrate that IR-induced senescent OCYs can secrete more CCL5, which in turn promotes osteoclastogenesis and enhances their bone resorption capacity. In order to explore the genesis and regulation mechanism of this inflammatory circumstance, the regulatory role of the JAK1/STAT3 pathway on CCL5 secretion was explored. The JAK1/STAT3 pathway is a stress-induced inflammatory signaling pathway that cascades to amplify the production and secretion of key SASP factors. In addition, the JAK1/STAT3 pathway may be activated by chemokines or cytokines, such as CCL5 [22,23,24]. We speculated that the JAK1/STAT3 pathway was involved in regulating the initiation of inflammation through CCL5. In this study, the irradiated OCYs were treated with a neutralizing antibody to CCL5. It was found that the protein expression level of p-JAK1 and p-STAT3, but not of t-STAT3, was elevated in the irradiated OCYs. Interestingly, the neutralizing antibody to CCL5 modestly reduced the IR-induced p-STAT3 levels, although the expression of t-STAT3 was unaffected (Figure 6A; Appendix A). The effect of CCL5 on the nuclear translocation of STAT3 was further verified using immunofluorescence to observe the nuclear translocation, which plays a crucial role in transcription factor function. The p-STAT3 protein was upregulated in the nucleus of the irradiated OCYs, while the neutralizing antibody to CCL5 inhibited p-STAT3 nuclear translocation in irradiated OCYs (Figure 6A). Meanwhile, the ELISA results indicated that the neutralizing antibody to CCL5 could silence the inflammatory milieu and reduce the levels of CCL5 in OCY supernatants by 21.40% compared with the irradiated OCYs (Figure 6B). Since the inhibition of CCL5 signaling by neutralizing antibodies was shown to significantly reduce osteoclastogenesis, we further investigated the mechanism of this osteoprotective effect. It was observed that the functional marker RANKL expression was upregulated in irradiated OCYs, whereas OPG expression was downregulated significantly, while the neutralizing antibody to CCL5 reversed the expression of RANKL and OPG (Figure 6C), suggesting that CCL5 activates IR-induced osteoclastogenesis by regulating the JAK1/STAT3 pathway.

## 3. Discussion

Cancer-treatment-induced bone loss (CTIBL) is a long-term complication associated with cancer treatment that can affect bone metabolism directly or indirectly, thus posing a great challenge to bone health and structural integrity and causing complications such as pathologic fractures, bone atrophy, and osteonecrosis [6]. Therefore, an in-depth exploration of the molecular mechanisms of radiological bone injury is needed to provide a foundation for intervention targets to alleviate bone damage after radiotherapy. In mature bone tissue, OCYs account for about 90% to 95% of all bone tissue cells, and they are also the longest-living cells in bone [25,26]. However, previous research on bone tissue has primarily focused on OBs due to the fact that OCYs are deeply embedded in hard bone tissue and extremely difficult to isolate and extract [27]. This article describes the initial isolation and culture of primary OCYs, where the identification of primary OCYs was based on cell morphology with slender synapses, E11-specific protein expression, and weak ALP expression.

Due to its high calcium content, bone can absorb significantly more radiation, up to 30% to 40%, compared to surrounding tissues [8,17]. As OCYs are the predominant cells in mature bone tissue, the direct damage to OCY biological function caused by IR is primarily due to the alteration in microcirculation, leading to cellular apoptosis, morphology damage, and functional impairment, which ultimately diminishes the ability to dynamically deposit surrounding organic and mineral material, as well as remodel the surrounding bone matrix. Actin filaments, in particular, are critical in maintaining the shape of the primary OCYs. The irradiated OCYs clearly showed a significant shortening in dendrite length, disorganized cytoplasm, and enlarged cytosol, showing a tendency to lose their dendritic branching structure, which was consistent with the reduced expression of E11, implying that IR disrupted OCYs’ synaptic structure and weakened their ability to join, segregate, and re-join, leading to the dysfunction of direct communication between the embedded OCYs and surrounding histocytes. In addition, OCYs are multifunctional cells that, in addition to sensing the global and local microenvironment, may also influence distant organ cells through the secretion of paracrine and endocrine factors [28,29]. In this study, we demonstrated that IR negatively regulated the expression of DMP-1, which is related to OCY mineralization and maturation. Furthermore, this study revealed that IR significantly upregulated RANKL expression, which is a cytokine that promotes OC differentiation, whereas OPG, an inhibitor of RANKL, was significantly downregulated, resulting in an elevated RANKL/OPG ratio. These changes represent an imbalance in the local microenvironment, with an increase in OC production, which in turn results in excessive bone resorption and a negative impact on bone mineral density. The above findings demonstrate that IR can disrupt bone homeostasis by inducing decreased OCY viability, morphologic damage, and functional impairment.

After exposure to IR, there are large clusters of DSBs in the cell, followed by a series of stress responses [30,31]. DSBs promote the recruitment and binding of kinases from ATM to sites of DNA damage, thereby driving the phosphorylation of histone H2AX to form γ-H2AX, which in turn facilitates the assembly of specific DNA repair complexes [32]. In the present study, the increased accumulation of γ-H2AX of irradiated OCYs suggested that IR caused double-strand breaks in DNA. Prior studies have demonstrated that temporally unrepaired DSBs can lead to termination segregation, and the end misconnection of these broken DNAs usually gives rise to chromosomal aberrations [31]. This study further revealed a marked increase in the characteristic heterochromatin structures in the nuclei of irradiated OCYs as punctate aggregates. SAHF is a special type of heterochromatin with a very stable structure and isolates many genes wrapped around chromatin [32]. The role of SAHF is believed to be the suppression of the transcription of genes associated with cellular proliferation. Therefore, once SAHF is formed, the cell irreversibly loses the ability to replicate and proliferate DNA and is predictive of the onset of cell senescence [32]. Furthermore, as the most widely used assay for senescent cells, SA-β-gal is a major glycoside hydrolase and glycosyltransferase, and its activity is used as a surrogate marker for the increase in lysosomal content in senescent cells. X-Gal produces a dark blue product when it is catalyzed by β-galactosidase specific for cellular senescence [33]. In this study, a significant increase in the number of cells stained positive for β-galactosidase was observed in irradiated OCYs, suggesting that senescence-related pathophysiological changes occurred. As is well known, the p21 and p53 proteins can constitute a G1 cell cycle checkpoint, checking for sites of DNA damage during the G1 phase and monitoring genomic integrity. Concurrently, the p16 protein inhibits the activity of CDK4 and ultimately prevents cells from entering the S phase, resulting in cell cycle arrest and senescence [34,35,36]. In the present study, the activation of senescence-related protein pathways p16 and p21 was observed in irradiated OCYs, suggesting that IR is sufficient to activate both the p21-Rb and p16-Rb signaling pathways, thus inducing cellular senescence. This result is in agreement with previous studies and suggests that 2 Gy IR of OCYs can successfully construct a cellular senescence model in vitro.

It has been shown that senescent OCYs and their SASP contribute to age-associated bone loss. Similar to aging, IR accelerates OCY senescence and SASP secretion and further promotes bone resorption. In short, SASP may in part account for the effects of senescent cells on bone homeostasis. The ability to target the SASP secreted by senescent cells to slow bone degeneration has now been reported in the literature. It has been reported that TNF inhibition brings about a balance between the action of OCYs on OCs. In addition, the inhibition of IL-6/IL-6R has also been shown to attenuate osteoclastogenesis, which contributes to our understanding of the role of inflammatory factors in the interaction between OCYs and the precursors of OCs [37,38]. These findings validate the belief that the pharmacological interventions targeting the SASP profile could be an effective strategy for improving bone integrity. In the present study, in addition to direct functional damage to OCYs, senescent OCYs induced by IR significantly upregulate SASP secretion, with CCL5 being one of the most abundant factors secreted. CCL5 has been shown to bind with a high specificity to CCR5, and upon activation, it forms the CCL5/CCR5 biological axis, which can signal intracellularly and plays a key role in the inflammatory process. 

The direct effects of ionizing radiation on osteoclasts have been studied. Specifically, ionizing radiation has a bidirectional regulatory effect on osteoclasts, manifested by the promotion of osteoclast differentiation and function at low doses of ionizing radiation, while high doses of ionizing radiation are mainly manifested by the inhibition of osteoclastogenesis [39,40]. In this study, we focused on whether ionizing radiation affects OC metabolism indirectly through cellular molecular coupling. It is widely acknowledged that senescent OCYs can secrete an array of soluble cytokines that accelerate the process of bone injury by activating osteoclastogenesis. Furthermore, it has been shown that when OCYs receive different doses of radiation, the levels of CCL5 secreted by OCYs are dependent on radiation dose [17,41]. To verify the paracrine regulation on osteoclastogenesis by IR-induced chemokine CCL5, the CM of OCYs was co-cultured with RAW264.7 precursor cells. The results showed that in the irradiated group, osteoclast precursors had an enhanced migration ability and mature intracellular microfilaments; the expression of function-specific *TRAP*, *CTSK*, *Oscar*, *Ocstamp*, and integrin *αV*, *β3* were markedly upregulated; and bone destruction was active. The neutralizing antibody to CCL5 disrupted the actin ring, forming OCs with dendritic pseudopodia and a disorganized cytoskeletal structure, reducing the expression of function-specific genes, and silencing bone destruction. Furthermore, in our experiments, it was observed that the neutralizing antibody to CCL5 significantly reversed the expression of RANKL/OPG, which ulteriorly reduced the expression of *TRAP*, *CTSK*, *Oscar*, *Ocstamp*, and integrin *αV*, *β3*. These findings are in agreement with those of Feng et al., who found that both MIP-1β and CCL5 promoted the migration of macrophages in a concentration-dependent manner, while CCL5 promoted OC formation [21]. According to previous research, the increase in serum CCL5 levels may specifically reflect the rate of bone metabolism in osteoporosis-prone individuals [21]. In addition, concerning pathological bone destruction, some observations report that the pro-osteogenic effect of CCL5 can be partially blocked with the blockade of the CCR5 receptor [42]. These results reinforce our findings. Our experiments further refine the notion that IR stimulates SASP secretion from senescent OCYs, particularly the elevated secretion of CCL5, which promotes precursor OC cell migration and fusion, further accelerating OC formation and maturation, and ultimately consolidating the functional onset of OCs. In conclusion, it remains to be seen whether targeting CCL5 is a promising strategy to alleviate IR-induced senile bone damage, which warrants further investigation.

In the present study, we focused on the JAK1/STAT3 pathway as a stress-induced inflammatory signaling pathway that cascades to amplify the genesis and secretion of key SASP factors. Previous studies have demonstrated that STAT3 phosphorylation is constitutively activated and retained upon the stimulation of CCL5 [22]. In light of these reports, it was found that the level of the phosphorylation of STAT3, which plays an important role in inflammatory factor storm, was significantly activated upon the stimulation of CCL5, a finding which is in agreement with previous reports [22,43,44,45,46]. Notably, we observed that the neutralizing antibody to CCL5 attenuated p-STAT3 expression, blocked the positive feedback mechanism of the pathway, and ultimately led to the less pronounced upregulation of the expression of the chemokine CCL5, which is in agreement with existing reports [47]. It was concluded that CCL5 activates IR-induced inflammation through the regulation of the JAK1/STAT3 pathway, which establishes a solid theoretical basis for the development of JAK1/STAT3 signaling as a potential therapeutic target for the inhibition of CCL5 secretion. In the future, a therapeutic strategy aimed at blocking CCL5 secretion from senescent OCYs by interfering with JAK1/STAT3 signaling may be a powerful approach to mitigate IR-induced senile bone damage.

## 4. Materials and Methods

### 4.1. Isolation of Primary OCYs

Primary OCYs were isolated from the femur and tibia of BALB/c mice following a protocol previously established in our laboratory [8,17]. All experimental procedures were approved by the Committee for Ethical Use of Experimental Animals at Fudan University (approval number: 2020-04-FYS-ZGY-01). In brief, BALB/c mice (male, 6 weeks old, weight 20 ± 2 g) were purchased from JieSijie Laboratory Animal Co., Ltd. (Shanghai, China). The calvariae were dissected and followed by six sequential 30-min digestions with 0.25% Trypsin (Gibco, Eggenstein, Germany), 0.001 g/mL collagenase type-II (Sigma-Aldrich, St. Louis, MO, USA), EDTA solution, and 0.001 g/mL collagenase type-I (Sigma-Aldrich, St. Louis, MO, USA) at 37 °C. The remaining bone fragments were collected and cultured at 37 °C in 5% CO_2_.

### 4.2. Cell Culture and IR Treatment

All cells were inoculated and cultured on plates coated with rat tail collagen type-I (Solarbio, Beijing, China) in an α-minimum essential medium (α-MEM; Gibco, Eggenstein, Germany) supplemented with 10% FBS (Gibco, Eggenstein, Germany) and 1% penicillin–streptomycin (PS; Sigma-Aldrich, St. Louis, MO, USA) in a humidified incubator at 37 °C with 5% CO_2_. When cells were 70% confluent, OCYs were exposed to 2 Gy using an X-Rad 320 Biological Irradiator (X-RAD 320, PXi). The dose rate was 100 cGy/min, with 320 kV tube voltage, 4.2 mA tube current, and a distance of 55 cm between the source and the surface. The OCYs of the control group were irradiated with 0 Gy. The cells continued to be cultured post-irradiation, and the medium was changed every other day. 

RAW264.7 precursor cells were recovered from cryopreservation, regularly tested for mycoplasma contamination, and incubated in plates with α-MEM medium containing 50 ng/mL RANKL.

### 4.3. IR-Induced Morphological and Functional Changes in OCYs

Cell viability was determined using the CCK-8 assay (C0037, Beyotime Institute of Biotechnology, Shanghai, China). Briefly, OCYs were seeded in 96-well plates (3 × 10^3^ cells/well) and then exposed to IR the next day, where they continued to incubate for 3 days. Subsequently, 10 µL CCK-8 solution was added into each well for 4 h at 37 °C, and the absorbance was measured at a wavelength of 450 nm using a microplate reader (Biotech, Vicenza, Italy).

Immunofluorescence analysis: Briefly, OCYs were inoculated on 3.5 cm OCY-specific crawlers placed in 6-well plates (1 × 10^5^ cells/well). The cells were irradiated and incubated until 50% confluence was reached, and then they were fixed with 2.5% glutaraldehyde, incubated for 1 h with 200 μL of tetramethyl rhodamine–phalloidin (Solarbio, Beijing, China) working solution (1:500 dilution) in the absence of light, and re-stained with 200 μL of 4′,6-diamino-2-phenylindole (DAPI; Dojindo, Kumamoto, Japan) solution (1:500 dilution) for 30 s. Images were taken using a Leica fluorescent microscope (Leica Microsystems, Wetzlar, Germany) with selected TRITC excitation–emission filters (Ex/Em = 540/570 nm) and DAPI excitation–emission filters (Ex/Em = 364/454 nm) with a magnification of ×200, and the dendrite length of the OCYs was measured using SimplePCI 6.6 software (C-imaging, Lake Oswego, OR, USA).

### 4.4. IR-Induced OCY Senescence and Secretory Phenotype

SA-β-gal staining: Briefly, OCYs were seeded in 6-well plates (1 × 10^5^ cells/well) and then exposed to IR the next day. The cells were incubated for 48 h, washed with PBS, and fixed with the fixative solution for 15 min. Then, the cells were incubated with freshly prepared senescence β-galactosidase staining solution (C0602; Beyotime Institute of Biotechnology, Shanghai, China) overnight at 37 °C without CO_2_. Finally, green-stained positive cells were photographed and counted from ten different regions per well using a light microscope at ×100 magnification.

SAHF detection: SAHF, visualized as DAPI-dense foci, is a feature of senescent cells. To identify the SAHF, OCYs were fixed with 4% paraformaldehyde for 15 min. Then, the cells were stained with DAPI for 5 min. DAPI-stained nuclei with blue fluorescence were finally photographed and counted from ten different regions per well under a Leica fluorescent microscope with a magnification of ×200.

IR-induced DNA damage: OCYs were fixed with 4% cold paraformaldehyde for 15 min and treated with 0.5% Triton-X100 for 10 min at room temperature. The cells were then analyzed using antibodies against γ-H2AX (ab81299; 1:250), CCL5/Rantes (2989; 1:250), and p-STAT3 (AF3293; 1:250) overnight at 4 °C in a wet box. Subsequently, the cells were incubated with anti-mouse- or anti-rabbit-IgG-HRP-conjugated secondary antibodies for 2 h in the dark, and nuclei were stained with DAPI for 5 min. γ-H2AX^+^ cells, CCL5^+^ cells, and p-STAT3 cells were finally photographed and counted using a Leica fluorescent microscope with a magnification of ×200.

Cytokine antibody microarray for multiple SASP: In total, 100 μL blocking solution was added to each well and blocked for 30 min. Then, 100 μL of the sample was added to each well and incubated overnight. Following this, 70 μL of biotin-labeled antibody was added and incubated for 1 h, and 70 μL of fluorescein-labeled streptavidin anti-biotin protein was added to each well and incubated for 1–2 h in the absence of light at room temperature. Signals were detected using chemiluminescence (ChemiScope 6300, CLINX, Shanghai, China) and subsequently quantitated with HLImage++ computer vision systems (Version 1.0.0.1, Western Vision Software, Salt Lake City, UT, USA).

Enzyme-linked immunosorbent assay for key SASP: The supernatant of OCYs was collected and subjected to an ELISA according to the manufacturer’s instructions. To the wells of the ELISA plate, we added 10 μL of the sample and 40 μL of the sample dilution. A total of 100 μL of ELISA reagent was added to each well, and then the plate was sealed with sealing film and incubated at 37 °C for 60 min. Subsequently, 50 μL of chromogenic agent A and 50 μL of chromogenic agent B were added to each well, and the wells were gently shaken and mixed for 15 min in the absence of light at 37 °C. The absorbance (OD value) of each well was measured sequentially at a 450 nm wavelength.

### 4.5. IR-Induced Senescent-OCY-Mediated Active Osteoclastogenesis

The collection of irradiated OCYs conditioned medium: OCYs were seeded in a T25 flask and then exposed to 2 Gy IR the next day. The cells were incubated for 48 h, and then a serum-free medium was replaced, where the cells continued to incubate for 24 h. Subsequently, the cell culture supernatant was collected and filtered with a 0.22 μm suction filter and termed the conditioned medium (CM). The cell culture supernatants from unirradiated and irradiated OCYs were labeled as CM-0 Gy and CM-2 Gy, respectively. Furthermore, the irradiated OCYs were treated with 0.5 μg/mL of CCL5 neutralizing antibody, and the cell culture supernatants were labeled as CM-2 Gy + anti-CCL5 according to the same procedure described above.

Cell migration assay for osteoclastogenesis: The migration process was observed in the scratch wound model and in the Transwell migration assay model in vitro. For the wound-healing assay, RAW264.7 precursor cells (4 × 10^5^ cells/well) were plated on 6-well dishes with 2 mL of collected CM and grown to confluence. After treating the cells with mitomycin (1 μg/mL) for 1 h, scratches were made using a p200 pipette tip. Scratch healing was observed at 24 h under a light microscope at ×20 magnification, and the percentage of wound closure was quantified using ImageJ. The wound-healing rate was calculated as (wound area after scratch − wound area after incubation)/(wound area after scratch) × 100%. For the Transwell migration assay, RAW264.7 precursor cells (5 × 10^4^ cells/well) in 200 μL of serum-free α-MEM were plated in the upper chambers, and 800 μL of the collected CM was added in the lower chambers. After 48 h, the cells on the lower chamber, which migrated from the upper chamber, were fixed with methanol and stained with 0.5% crystal violet. Ten randomly selected images per well were acquired under a light microscope at ×100 magnification, and the average number of stained cells was determined using ImageJ.

Tartrate-resistant acid phosphatase staining (TRAP) assay for osteoclastogenesis: RAW264.7 precursor cells (4 × 10^3^ cells/well) were plated in a 24-well cell culture dish for 6 days, with collected CM supplemented with 50 ng/mL RANKL. Adherent cells were fixed and stained in a TRAP Kit (387A-1KT, Sigma-Aldrich, St. Louis, MO, USA) according to the manufacturer’s protocol. TRAP^+^ multinucleated (3–6 nuclei) cells were identified as small OCs, TRAP^+^ multinucleated (6–10 nuclei) cells as medium OCs, and TRAP^+^ multinucleated (>10 nuclei) cells as large OCs. Meanwhile, the number of three types of cells was quantified using ImageJ.

### 4.6. Western Blot Analysis

Primary OCYs were lysed in RIPA lysis buffer (Beyotime Institute of Biotechnology, Shanghai, China) at 4 °C. The protein concentration was quantified using the BCA Protein Assay Kit (Beyotime Institute of Biotechnology, Shanghai, China). Total protein (20 μg) was separated via 10% SDS-PAGE (PG113; Epizyme Biotech, Shanghai, China) and transferred onto 0.45 µm PVDF membranes (Millipore, Billerica, MA, USA). After blocking with 5% skimmed milk in TBS supplemented with 0.1% Tween 20 (TBST) for 1 h, the membranes were incubated with a primary antibody at 4 °C overnight. Subsequently, membranes were washed and incubated with an anti-mouse-IgG-HRP-conjugated or anti-rabbit-IgG-HRP-conjugated secondary antibody (SA00001-2; 1:1000) for 1 h at room temperature. Finally, immunoreactive bands were detected with an ECL Kit (D3308; Beyotime Institute of Biotechnology, Shanghai, China) and an Omega Lum™ C Imaging System (Gel Company, San Francisco, CA, USA). Densitometry analysis was performed using ImageJ 1.8.0 Software (National Institutes of Health, Bethesda, MD, USA).

Western blotting was performed using total protein extracts and probing with antibodies against β-actin (4970S; 1:1000), p16 (ab51243; 1:1000), p21 (ab188224; 1:1000), γ-H2AX (ab81299; 1:1000), E11 (df6824; 1:1000), RANKL (ab45039; 1:1000), OPG (ab183910; 1:1000), DMP-1 (ab177246; 1:1000), CCL5/Rantes (2989; 1:1000), STAT3 (9139T; 1:1000), p-STAT3 (9145T; 1:1000), and p-JAK1 (AF2012; 1:1000). 

The original blots with molecular marker positions in the paper are shown in Appendix A. The Appendix A of the paper are shown in Appendix A.

### 4.7. RNA Extraction and Reverse-Transcription Quantitative PCR (RT-qPCR)

Total RNA was isolated from OCYs using TRIzol^®^ reagent (BSC52S1; Bioflflux, Beijing, China). Then, cDNA was synthesized using a High-Capacity cDNA Reverse Transcription Kit (KR118-02; Tiangen Biotech, Beijing, China), and RT-qPCR reactions were performed using Power Up SYBR-Green Master Mix (Invitrogen; Thermo Fisher Scientific, Inc., Waltham, MA, USA) with the ABI QuantStudio 5 (Applied Biosystems, Carlsbad, CA, USA). *GAPDH* was used as the loading control. The PCR for each sample was conducted in triplicate. The primers were designed using GenBank sequences and are listed in Table 1.

### 4.8. Statistical Analysis

All experiments were performed at least twice. Data were presented as the mean ± SD, and calculations were analyzed using SPSS 20.0 software (SPSS Inc., Chicago, IL, USA). T-tests were performed between two groups, one-way ANOVAs were performed between multiple groups, and post hoc Bonferroni tests were used for multiple comparisons. A value of * *p* < 0.05 was considered statistically significant.

## 5. Conclusions

In summary, the present study reveals that CCL5 reinforces the IR-induced inflammatory process and osteoclastogenesis through the activation of the JAK1/STAT3 pathway, which strongly suggests that CCL5 is a potential molecular target for the treatment of IR-induced senescent-based bone loss. Simultaneously, animal experiments were carried out in vivo to better understand the specific mechanisms and responses of osteocytes to irradiation exposure. However, our study does have some limitations. Thus, to further elucidate the specific molecular mechanisms through which CCL5 supports osteoclastogenesis, more in-depth studies should be conducted in this research area. Furthermore, in addition to CCL5, as a key factor in SASP secretion by senescent OCYs induced by IR, a few important factors remain to be uncovered in the problem of IR-induced senile bone damage.

## Figures and Tables

**Figure 1 ijms-24-16168-f001:**
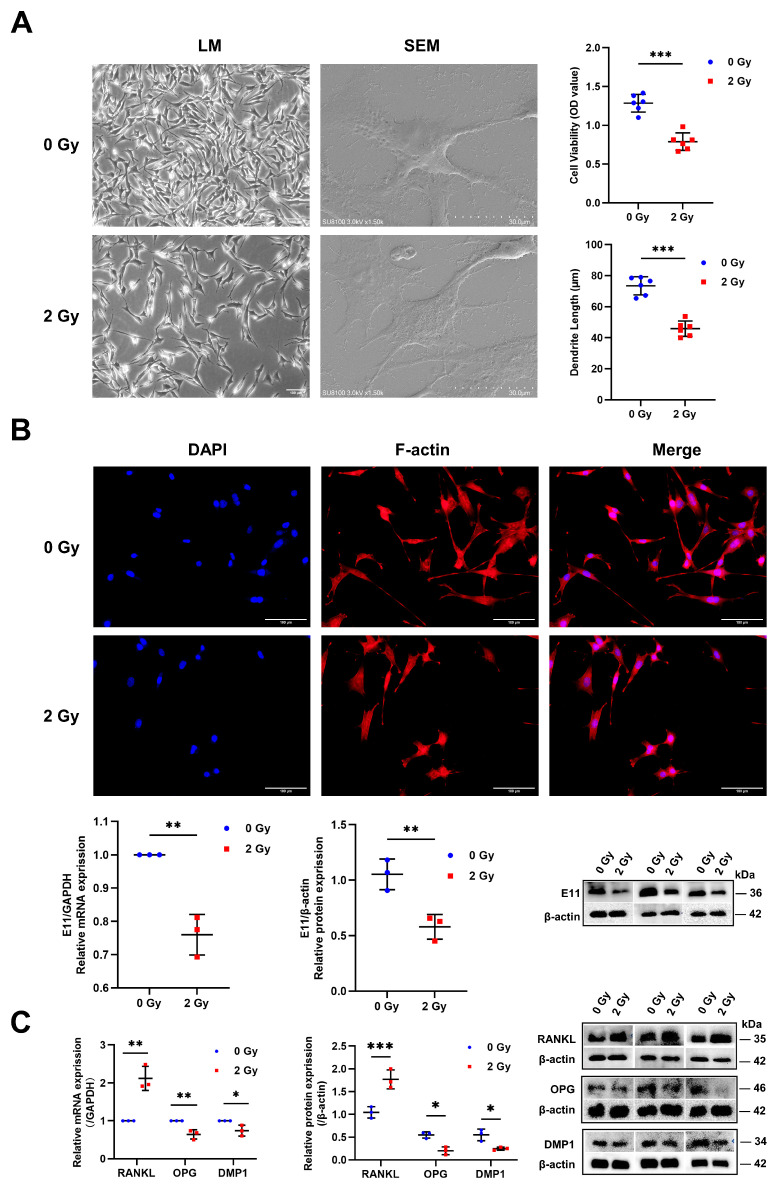
Radiation impairs the morphology and function of OCYs: (**A**) Representative image of primary OCY morphology via LM and SEM at 3 days post-irradiation; magnification = ×100 and ×1.5 k. Variation in OCY viability by CCK-8; *n* = 6. Quantification of dendrite lengths per cell; *n* = 6. (**B**) Immunofluorescence staining of OCY using phalloidin-AlexaFluor488 to visualize the typical dendrite-like synapse (shown in red) and DAPI fluorescence to visualize the nuclei (shown in blue); magnification = ×200. Protein and relative mRNA expression levels of the classical markers E11 in OCYs were studied via Western blot and RT-qPCR analysis; *n* = 3. (**C**) Protein and relative mRNA expression levels of the functional markers RANKL, OPG, and DMP-1 in OCYs; *n* = 3. Results are expressed as mean ± SD (* *p* < 0.05; ** *p* < 0.01; *** *p* < 0.001 vs. 0 Gy). LM, light microscopy; SEM, scanning electron microscopy; CCK8, cell counting kit-8; OPG, osteoprotegerin; RANKL, receptor activator for nuclear factor-kB ligand; DMP-1, dentin matrix acidic phosphoprotein l.

**Figure 2 ijms-24-16168-f002:**
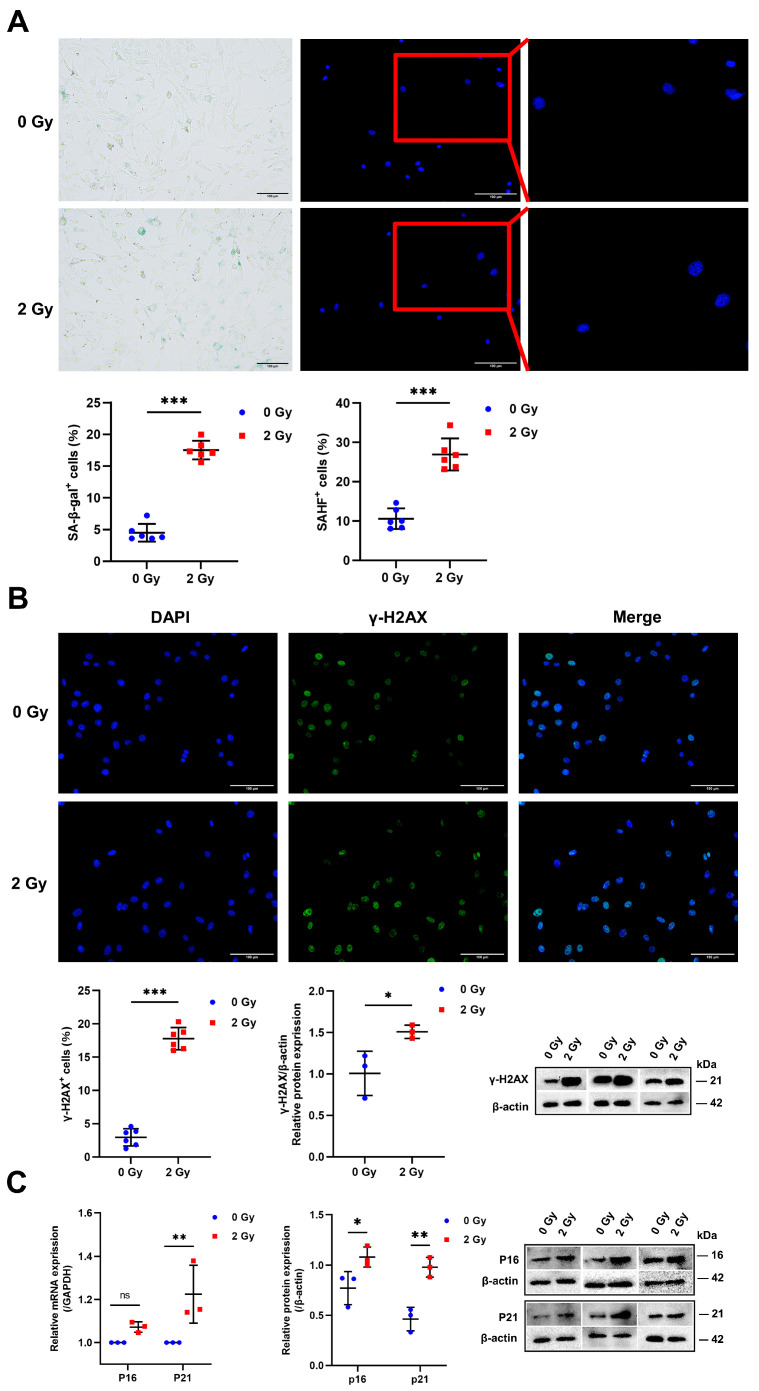
Radiation initiates cellular senescence in OCYs: (**A**) Representative images of SA-β-gal staining in primary OCYs post-irradiation; magnification = ×100. Quantification of SA-β-gal^+^ cells; *n* = 6. Representative images of SAHF formation in OCYs nuclei via FM; magnification = ×200. Quantification of SAHF^+^ cells; *n* = 6. (**B**) Immunofluorescence staining for γ-H2AX of OCYs: γ-H2AX (shown green) and DAPI (shown in blue); magnification = ×200. Quantification of γ-H2AX^+^ cells; *n* = 6. Protein expression levels of γ-H2AX in OCYs; *n* = 3. (**C**) Protein and relative mRNA expression levels of p16 and p21 in OCYs were studied via Western blot and RT-qPCR analysis, respectively; *n* = 3. Results are expressed as mean ± SD (ns, not significant; * *p* < 0.05; ** *p* < 0.01; *** *p* < 0.001 vs. 0 Gy). SA-β-gal, senescence-associated β-galactosidase; SAHF, senescence-associated heterochromatic foci; FM, fluorescence microscopy.

**Figure 3 ijms-24-16168-f003:**
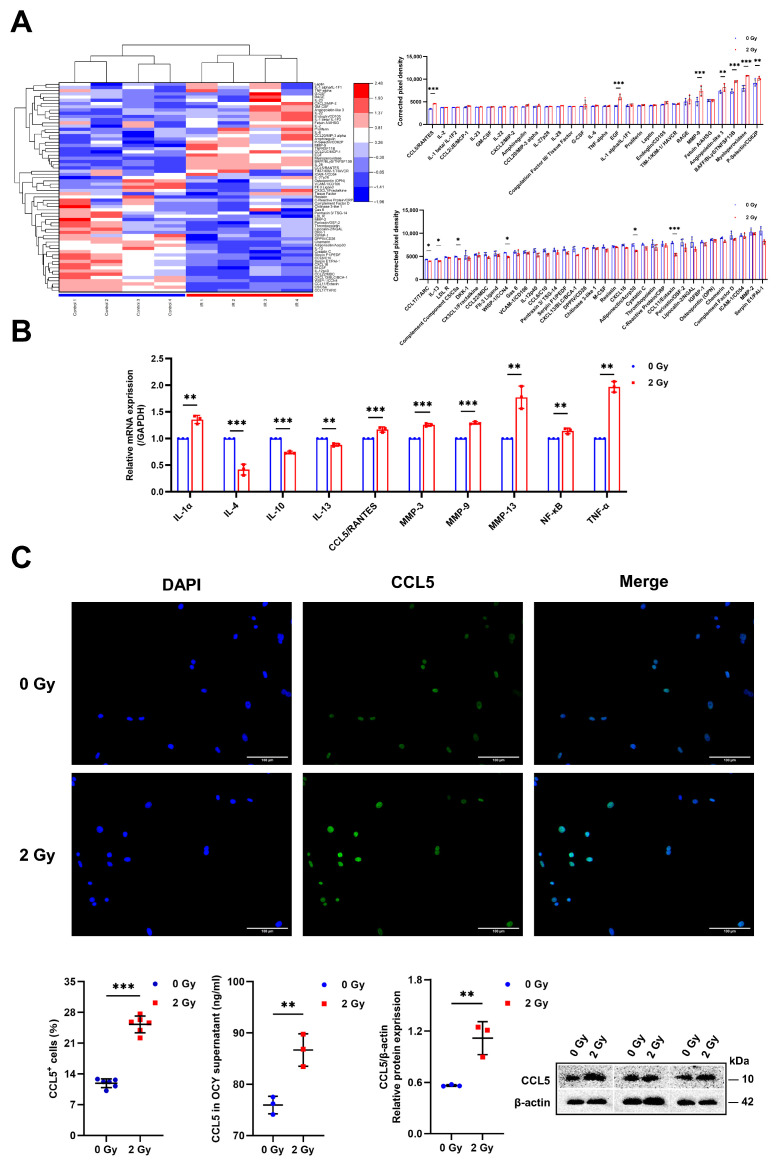
Radiation enhances SASP secretion in OCYs: (**A**) The heat map of the SASP composition with clustering or rows and columns in primary OCY serum post-irradiation via mouse cytokine microarray membrane analysis. Quantified expression levels of the OCY-specified secretory cytokines, expressed as the corrected pixel density; *n* = 4. (**B**) Relative mRNA expression levels of the OCY-specified secreted cytokines and chemokines were studied via RT-qPCR analysis; *n* = 3. (**C**) Immunofluorescence staining for CCL5 of OCYs: CCL5 (shown in green) and DAPI (shown in blue); magnification = ×200. Quantification of CCL5^+^ cells; *n* = 6. Expression levels of CCL5 in OCY supernatants were studied via ELISA assay; *n* = 3. Protein expression levels of CCL5 in OCYs were determined via Western blot analysis; *n* = 3. Results are expressed as mean ± SD (* *p* < 0.05; ** *p* < 0.01; *** *p* < 0.001 vs. 0 Gy). SASP, senescence-associated secretory phenotype; MMP-9, matrix metallopeptidase 9.

**Figure 4 ijms-24-16168-f004:**
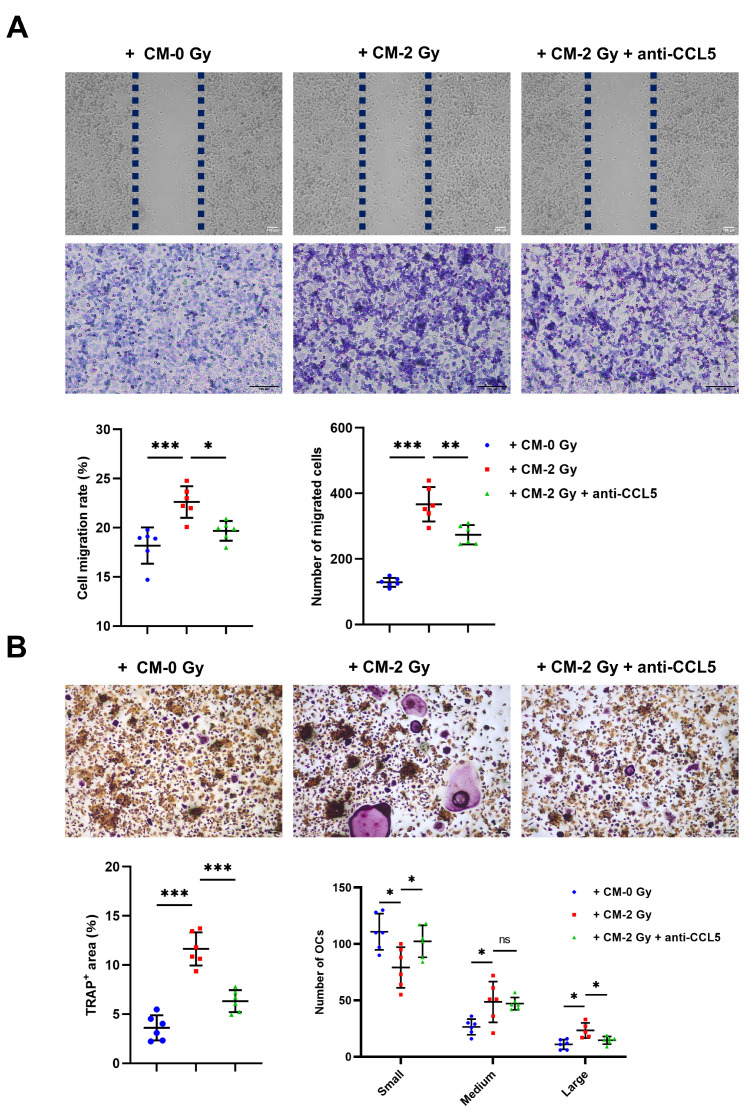
CCL5 accelerates the migration of RAW264.7 precursor cells: (**A**) Changes in the migration rate and migration numbers of RAW264.7 precursor cells co-cultured with CM-0 Gy or CM-2 Gy and CM-2 Gy + anti-CCL5 using a wound-healing assay and Transwell migration assay, respectively; magnification = ×20 and ×100. Quantification of the migration rate and the migration numbers of RAW264.7 precursor cells; *n* = 6. (**B**) Changes in the TRAP^+^ area and numbers of OCs co-cultured with CM-0 Gy or CM-2 Gy and CM-2 Gy + anti-CCL5 using TRAP staining; magnification = ×40. Quantification of TRAP^+^ area; *n* = 6. Quantification of the number of small OCs, medium OCs, and large OCs; *n* = 6. Results are expressed as mean ± SD (ns, not significant; * *p* < 0.05; ** *p* < 0.01; *** *p* < 0.001 vs. 0 Gy). CM-0 Gy, conditioned medium with no irradiation; CM-2 Gy, condition medium with 2 Gy X-ray; CM-2 Gy + anti-CCL5, condition medium with 2 Gy X-ray following CCL5 neutralizing antibody. TRAP^+^ multinucleated (3–6 nuclei) cells were identified as small OCs, TRAP^+^ multinucleated (6–10 nuclei) cells as medium OCs, and TRAP^+^ multinucleated (>10 nuclei) cells as large OCs. TRAP, tartrate-resistant acid phosphatase.

**Figure 5 ijms-24-16168-f005:**
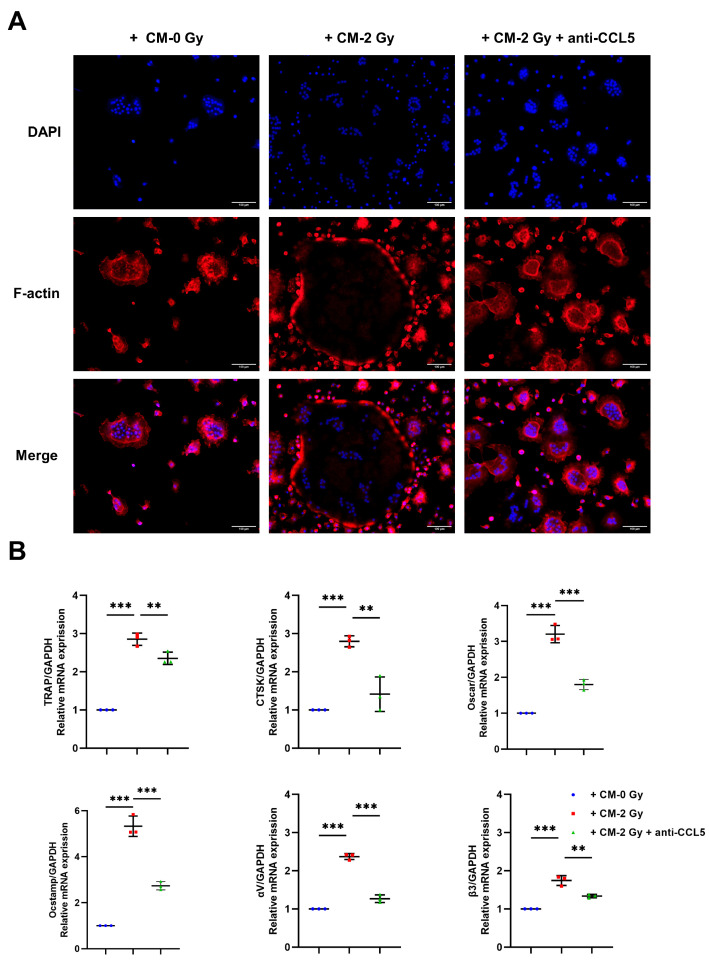
CCL5 promotes OC differentiation ability: (**A**) Immunofluorescence staining of the OC co-cultured with CM-0 Gy or CM-2 Gy and CM-2 Gy + anti-CCL5 using phalloidin-AlexaFluor488 to visualize the typical OC actin rings (shown in red) and DAPI fluorescence to visualize the nuclei (shown in blue); magnification = ×100. (**B**) Relative mRNA expression levels of OC functional markers *TRAP*, *CTSK*, *Oscar*, *Ocstamp*, and adhesion-related molecules *αv*, *β3* using RT-qPCR analysis; *n* = 3. Results are expressed as mean ± SD (** *p* < 0.01; *** *p* < 0.001 vs. 0 Gy). CTSK, cathepsin K; Oscar, osteoclast-related receptor; Ocstamp, osteoclast stimulatory transmembrane protein.

**Figure 6 ijms-24-16168-f006:**
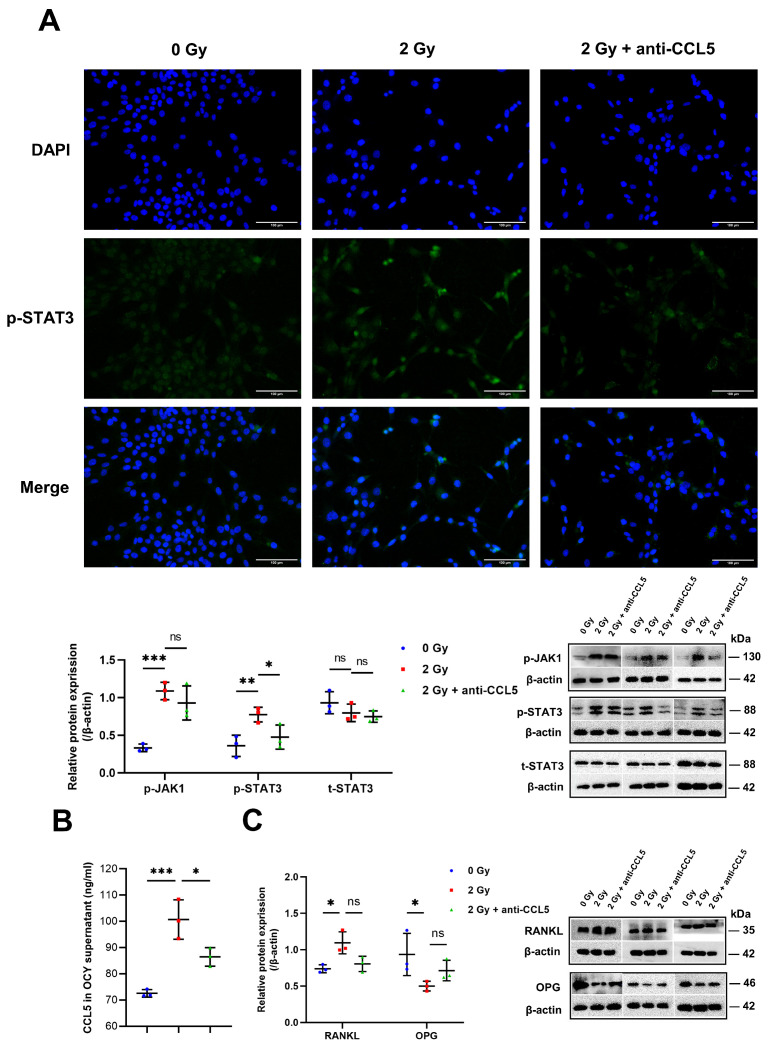
CCL5 activates the JAK1/STAT3 pathway: (**A**) Immunofluorescence staining for p-STAT3 of OCYs: p-STAT3 (shown in green) and DAPI (shown in blue); magnification = ×200. Protein expression levels of the functional markers p-JAK1, STAT3, and p-STAT3 in OCYs were assessed using Western blot analysis; *n* = 3. (**B**) Expression levels of CCL5 in OCY supernatants were assessed using ELISA assay; *n* = 3. (**C**) Protein expression levels of the functional markers RANKL, OPG in OCYs; *n* = 3. Results are expressed as mean ± SD (ns, not significant; * *p* < 0.05; ** *p* < 0.01; *** *p* < 0.001 vs. 0 Gy). JAK1/STAT3, Janus kinase1-signal transducer and activator of transcription 3.

**Table 1 ijms-24-16168-t001:** Primer sequences for quantitative reverse-transcription polymerase.

Target Genes	Primer Sequence
*GAPDH*	S 5′-AGGTCGGTGTGAACGGATTTG-3′
	A 5′-GGGGTCGTTGATGGCAACA-3′
*E11*	S 5′-ACCGTGCCAGTGTTGTTCTG-3′
	A 5′-AGCACCTGTGGTTGTTATTTTGT-3′
*RANKL*	S 5′-CGCTCTGTTCCTGTACTTTCG-3′
	A 5′-GAGTCCTGCAAATCTGCGTT-3′
*OPG*	S 5′-CCTTGCCCTGACCACTCTTAT-3′
	A 5′-CACACACTCGGTTGTGGGT-3′
*p16*	S 5′-CGCAGGTTCTTGGTCACTGT-3′
	A 5′-TGTTCACGAAAGCCAGAGCG-3′
*p21*	S 5′-CCTGGTGATGTCCGACCTG-3′
	A 5′-CCATGAGCGCATCGCAATC-3′
*TNF-α*	S 5′-TCAGAATGAGGCTGGATAAG-3′
	A 5′-GGAGGCAACAAGGTAGAG-3′
*NF-κB*	S 5′-TGCGATTCCGCTATAAATGCG-3′
	A 5′-ACAAGTTCATGTGGATGAGGC-3′
*IL-1α*	S 5′-CTGAAGAAGAGACGGCTGAGT-3′
	A 5′-CTGGTAGGTGTAAGGTGCTGAT-3′
*IL-10*	S 5′-GCTCTTACTGACTGGCATGAG-3′
	A 5′-CGCAGCTCTAGGAGCATGTG-3′
*IL-13*	S 5′-CCTGGCTCTTGCTTGCCTT-3′
	A 5′-GGTCTTGTGTGATGTTGCTCA-3′
*CCL5/RANTES*	S 5′-GCCCACGTCAAGGAGTATTTCT-3′
	A 5′-ACAAACACGACTGCAAGATTGG-3′
*MMP-3*	S 5′-AGGGATGATGATGCTGGTATG-3′
	A 5′-AACACCACACCTGGGCTTAT-3′
*MMP* *-* *9*	S 5′-CTGGACAGCCAGACACTAAAG-3′
	A 5′-CTCGCGGCAAGTCTTCAGAG-3′
*MMP* *-* *13*	S 5′-CCTTGATGCCATTACCAGTCTC-3′
	A 5′-TCCACATGGTTGGGAAGTTCT-3′
*Oscar*	S 5′-GCTATTACCACACGCCTTCTG-3′
	A 5′-CCAAGCAGATGAGGACCATTC-3′
*Ocstamp*	S 5′-AGCCACGGAACACCTCTT-3′
	A 5′-TGGGTCAGTAGTTCGTTACAG-3′
*T* *RAP*	S 5′-CACTCCCACCCTGAGATTTGT-3′
	A 5′-CATCGTCTGCACGGTTCTG-3′
*CTSK*	S 5′-CAGCAGAACGGAGGCATTGA-3′
	A 5′-CCTTTGCCGTGGCGTTATAC-3′
*αV*	S 5′-CTCAGGCTCTTCCACCACAT-3′
	A 5′-GACGGATACTGGCAAAAACG-3′
*β3*	S 5′-ATCCAAAATACGCAGCCATC-3′
	A 5′-GTTCCAAGAGCAGCAAGGAC-3′

## Data Availability

The datasets generated and analyzed in the present study are all included in this published article.

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
