# Peer review of "C-C Motif Chemokine Ligand 5 (CCL5) Promotes Irradiation-Evoked Osteoclastogenesis"

_ijms, 2023, doi:10.3390/ijms242216168_

Round 1
Reviewer 1 Report
Comments and Suggestions for Authors
The authors hypothesized that an increase in CCL5 among the SASP factors secreted by OCY activates IR-induced inflammation and osteoclastogenesis through the regulation of the JAK1/STAT3 pathway.
The authors make a description of the problem and establish the hypothesis.
Results are clear and can be easily followed
The discussion is adapted to the results obtained
The methodology is widely described and can be reproduced.
Reviewer 2 Report
Comments and Suggestions for Authors
The manuscript authored by Jing Wang et al, titled "CCL5 Mediates Osteoclastogenesis Targeting Inflammation of Irradiation-Induced Senescent Osteocytes," is both scientifically rigorous and well-articulated. The study's design is compelling and has been executed with meticulous care. To provide a stronger foundation for the study, the introduction should be refined to better highlight the key findings from prior studies, setting the stage for the rationale behind the present research. This will help readers to appreciate the significance and context of the current investigation more readily. I have the following concerns
1. It appears there might be a discrepancy between the information presented in the text (lines 101-104) and the data depicted in Figure 1C, specifically regarding the expression of DMP1. The text states that DMP1 is upregulated in irradiated OCYs, while Figure 1C suggests otherwise. This inconsistency should be addressed and clarified in the manuscript to ensure accurate reporting of the results. The authors may need to revise the text to accurately reflect the findings shown in the figure or provide an explanation for the apparent contradiction if it is due to an experimental condition or specific subset of samples.
2. It is a valuable suggestion to investigate the direct effects of ionizing radiation (IR) on osteoclastogenesis by studying the impact of IR on osteoclasts themselves. This would help in elucidating whether IR has a direct influence on osteoclast differentiation and function. The authors could consider conducting additional experiments to explore this aspect of their research. By doing so, they can provide a more comprehensive understanding of the mechanisms underlying the observed effects and potentially uncover new insights into the relationship between IR and osteoclast biology. This addition would contribute to the scientific rigor and completeness of their study.
3. Authors should conduct in vivo mouse models to replicate conditions more closely related to the human physiological response to IR. Conducting experiments where mice are irradiated, and osteocytes are subsequently isolated from bone tissue can provide valuable insights into the direct effects of ionizing radiation (IR) on osteocytes. This approach would allow for a more physiologically relevant investigation into how IR impacts these bone cells. By replicating experiments under these conditions, the authors can better understand the specific mechanisms and responses of osteocytes to IR exposure, which may contribute significantly to our knowledge of radiation-induced bone damage and inflammation.
Reviewer 3 Report
Comments and Suggestions for Authors
ijms-2632914
CCL5 mediates osteoclastogenesis targeting inflammation of irradiation-induced senescent osteocytes by Wang et al.
The manuscript describes in vitro experiments performed on primary murine calvarial osteocytes and RAW 264.7 macrophages.
Essentially the authors report radiation (2Gy) dependent alterations for ostocyte morphology, ratios of RNKL/OPG expression, indications for senescence and double strand breaks, CDKi expression (p16, p21) and secretome changes. CCL5 changes (1.28-fold mRNA, +13% protein) were studied further. RAW 264.7 cells co cultured with irradiated osteocytes showed increased motility and osteoclastic differentiation (TRAP, osteoclastogenic marker expression) mediated by STAT/JAK kinases.
Most of the described outcomes and signaling pathways suggested in the manuscript have been described before. The authors attempted to combine these findings to provide a functional relationship between ionizing irradiation and impaired bone remodeling.
In principle, this all sounds quite plausible, even if the biological relevance of a 1.28-fold increase (+13% at the protein level) of CCL5 in irradiated cells is debatable. However, it remains to be seen whether the study can show a clinical relevance as speculated.
In fact, radiotherapy has been used successfully for decades to treat bone metastases, and numerous studies have consistently shown that there is a transient decrease in bone density and thus bone remodeling activity, which is rapidly accompanied by increased bone formation and always an increase in bone density. This is independent of additional therapy with anti-resorptive drugs (bisphosphonates, denosumab, etc.). Therefore, it remains unclear to me whether the problem described here and used as a target actually exists clinically.
This should at least be described and discussed in detail in the Introduction and Discussion.
In the illustrations of the original Western blots and other available primary evidence, I would have expected the complete gels in each case and not processed in a similar way as they are already presented in the manuscript.
Overall, I believe that the clinical problem raised here does not exist at all, and therefore the study must be classified and discussed differently. In my opinion, major revisions are therefore necessary.
Comments on the Quality of English LanguageModerate editing of English language required
Round 2
Reviewer 2 Report
Comments and Suggestions for Authors
In future, the authors should study the mechanism thoroughly
Comments on the Quality of English LanguageMinor English editing required
Reviewer 3 Report
Comments and Suggestions for Authors
The authors have responded to the criticism. Clinically, the statement of the study remains questionable, but experimentally, the results seem authentic. Resolving this discrepancy must be the aim of future studies.
